# Intercultural Competencies for Fostering Technology-Mediated Collaboration in Developing Countries

Albert Kampermann [1,*], Raymond Opdenakker [1,2], Beatrice Van der Heijden [1,3,4,5,6,†] and Joost Bücker [3]

1. Faculty of Management, Open University of the Netherlands, P.O. Box 2960 Heerlen, The Netherlands; R.J.G.Opdenakker@tue.nl (R.O.); beatrice.vanderheijden@ru.nl (B.V.d.H.)
2. Sub Department Innovation, Technology Entrepreneurship & Marketing, Faculty Industrial Engineering & Innovation Sciences, Eindhoven University of Technology, P.O. Box 513 Eindhoven, The Netherlands
3. Institute for Management Research, Radboud University Nijmegen, P.O. Box 9108 Nijmegen, The Netherlands; joost.bucker@ru.nl
4. Department of Marketing, Innovation and Organization, Ghent University, 9000 Ghent, Belgium
5. Business School, Hubei University, Wuhan 430062, China
6. Kingston Business School, Kingston University, London KT2 7LB, UK
* Correspondence: albert.kampermann@ou.nl
† Main affiliation: Institute for Management Research, Radboud University Nijmegen, P.O. Box 9108 Nijmegen, The Netherlands is the main affiliation for Beatrice van der Heijden.

**Abstract:** With the rapid global spread and application of Information and Communication Technology (ICT), the question is whether every culture makes similar use of the ideology that often underlies its creators' design. ICT applications are designed with underlying beliefs or principles about e.g., work, communication, and individuality. These beliefs or principles are invisible and hidden in software and, as such, in many instances not recognized by users in other cultures. These hidden principles might even frustrate the understanding, use, knowledge-sharing, and e-collaboration between people from different cultures. In this article, we aim to explore, from a historical point of view, the early years of adaptation of ICT in developing countries, and we will highlight the importance of the use of intercultural (ICT-)skills to learn to recognize cultural differences from a relationship-based definition in technology-mediated collaboration. A semi-systematic or narrative review approach is used that is particularly suitable for topics that have been conceptualized differently. Our review firstly summarizes and categorizes the cultural factors impacting the adaptation and diffusion of ICT, especially in developing countries, and investigates which factors could hinder and/or facilitate the collaboration with other countries. Secondly, the findings of a thorough comparison between different intercultural competencies' frameworks indicate that intercultural competencies show a combination of motivation, knowledge (-management), and skills, which are key competencies in the light of successful technology-mediated collaboration.

**Keywords:** technology-mediated collaboration; e-collaboration; ICT adaptation; intercultural competencies; case examples; developing countries

## 1. Introduction

ICT is an important source of progress in collaborations between people of culturally different countries, especially between Western-oriented countries and developing countries. Collaboration refers to business-to-business collaboration, collaboration in the field of education, development cooperation, etc. In order to use ICT in this way, it is important to better understand in which way ICT and e-collaboration are perceived in developing countries? We use a semi-systematic or narrative review approach as a strategy to map different theoretical perspectives [1]. Our purpose of this review is not to cover all articles ever published on the topic but rather to combine different perspectives to understand the effect of cultural determinants in the use of ICT across countries. In general, ICT is seen as a key driver of economic development and as a means to reduce poverty in

developing countries [2], but there are major differences in the interpretation of the term developing countries. The designations "developed" and "developing" are intended for statistical convenience and do not necessarily express a judgement about the stage reached by a particular country or area in the development process [2]. These statistics include e.g., comparisons between income, economy, health, education, and safety. Earlier, ref. [3] stated that ICT is the most important factor separating the developing from the developed countries. We argue that the most common explanation or justification for the importance of ICT for developing countries lies in a classic, 'modernization' thesis [4], stating that ICT can foster services, such as health and education, and can support in the creation of economic opportunities [5–7]. Since developing countries may have fewer schools and teachers, doctors, and nurses, and a lower calorie intake per capita, in comparison with people in developed countries [8], ICT can play an important role, through its direct approach towards individuals in delivering up-to-date information and knowledge.

ICT refers to synchronized electronic communication media that enhance the exchange of information and knowledge, and thereby may create and facilitate opportunities for close(r) e-collaboration. For example, e-learning and e-collaboration are suitable ICT formats because of a range of benefits offered by the web-based environment for institutions from different countries [9,10].

First of all, it is cost-effective and easily available. E-collaboration facilitates time management; the world-wide web can be accessed anywhere, that is from any location. The on-line representation of the required material creates a kind of uniform learning environment, which according to [11], bridges differences in learning (styles) and therefore the gap between potentially uneven facilities in different institutions (equipment, technology, etc.).

However, an important misunderstanding, in this regard, is that capitalizing on the opportunities of ICT not only comprises the existence of infrastructure and access to the Internet, but also the existence of ICT-related human capacity. More specifically, Internet users would need to master different skills, such as technical, structural, and strategic ones, to be able to fully benefit from Internet access [12]. Technical skills comprise the skills that are needed for the practical use of ICT equipment, structural skills refer to understanding the underlying principles (how is it intended, what can I do with it?), and strategic skills denote the capacity to select information and opportunities to achieve certain goals with it [13].

Studying a wide variety of technologies, researchers typically focus on identifying enablers of and barriers to increasing the adoption and diffusion levels of ICT in developing countries [5]. Once barriers to adoption and diffusion of ICT are removed or reduced, the use of ICT for e.g., e-learning and e-collaboration, will be enhanced, and the expected economic opportunities will rise. We state that ICT and infrastructure that are easily adopted in Western countries, and that are strongly based on, and therefore also limited by, Western cultural social norms, ethical values, traditional customs, and belief systems' values, bias the use of ICT in developing countries. In this article, 'Western culture' refers to the heritage of political systems, artifacts, and technologies that originated in, or are associated with, Europe and countries and cultures that are strongly connected to Europe (e.g., USA, Australia) (Western culture (n.d.). In Wikipedia, from https://en.wikipedia.org/wiki/Western_culture (accessed on 7 August 2019).

We posit that there is an urgent need to accompany ICT and infrastructure by human capacity, more specifically, through intercultural competencies, as there is concern about Western cultural hegemony regarding 'the information society' over other world regions [14]. Basically, intercultural competencies include awareness, knowledge, skills, practices, and processes that people need in order to function effectively and appropriately in culturally-diverse situations, and to interact with people from different cultures [15]. Caligiuri [16] defined these competencies as knowledge, skills, abilities, and other personality characteristics (i.e., the KSAOs). Arasaratnam [17] stated that our understanding of 'intercultural' needs to be a subject of continual study as our identity and values are shaped more and more by multiple cultural influences. In the context of this article, we

posit that intercultural competencies are relevant for the understanding and fostering of technology-mediated collaboration.

Over the last decades, one important factor in the thinking about development in countries has been the 'accessibility to new technology' [18]. Up to now, most of the scholarly literature in this field is related to adoption and diffusion of (the use of) ICT in developing countries, which are more or less induced by foreign interventions. However, the cultural capacity to embed ICT locally in a certain society and its organizations is somewhat ignored. The importance of (cultural) context was already mentioned in a recent publication by [5], who referred to unwarranted or simplified assumptions about the benefits of ICT across the globe, without carefully fine-tuning its embedding, see e.g., Avgerou [19]. In a similar vein, Pradhan [18] notified some kind of common belief that ICT is 'good' for use in all countries and is a logical step in the progression of a society. However, before an elaborate implementation of ICT in developing countries can be undertaken, one should consider what role ICT plays in the development processes of these countries and how its role is affected by national culture?

Osterwalder [6] found that ICT-related capacities can be divided into three main types: (1) infrastructure-related capacities; (2) sector application-related capacities; and (3) user-related capacities (See Figure 1). First, ICT can only flourish in case there is sufficient capacity to provide and maintain ICT infrastructure at a reasonable price in a sustainable way. Second, ICT can only become valuable to people when useful local content is available. Therefore, a new class of entrepreneurs is needed that focuses upon developing the capacity to imagine, create, and maintain useful applications in different sectors that make sense to the local community. Finally, users should develop the capacity to understand and to use ICT [6]. More specifically, as latecomers to the 'ICT scene', developing countries are facing enormous difficulties of which perhaps the most important being that they are becoming users of ICT without having built up the necessary capacities [20–22].

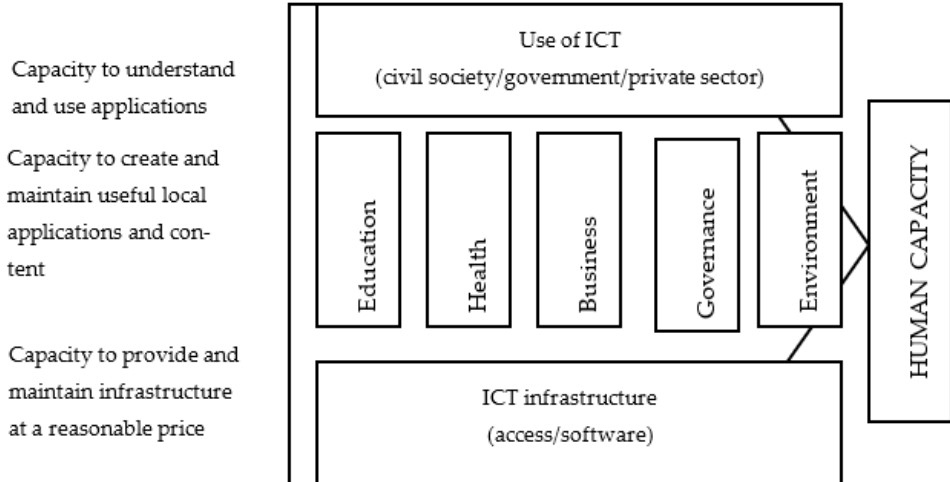

**Figure 1.** Cross-sectoral impact of ICT [6].

Basically, the main concern in this article is an exploration of the meaning of ICT in developing countries, specifically, the importance of 'making sense', and the understanding of the role and effects of ICT in communication and cooperation between countries that have both different national cultures and different stages of economic development. Despite its potential benefits, e-collaboration may present several challenges for participants as well. In this regard, ref. [23] referred to the influence of psychological collectivism in e-collaboration, deriving from different preferences, reliance, acceptance, and concern for one's own national cultural context. As the meaning of ICT is considered differently throughout (developing) countries, we should focus on a better understanding of the consequences thereof regarding in the light of e-collaboration with(in) these countries, and

how one, from a Western perspective, is able to collaborate with governments and NGOs in developing countries through the means of ICT.

Although we are aware of the importance of the earlier mentioned types of capacities as distinguished by [6]: (1) infrastructure-related capacities, (2) sector application-related capacities, and (3) user-related capacities; see Figure 1), in this article, we will explore the capacity of users to comprehend the benefits and therefore to use ICT applications. As such, we are mainly interested in the psychological and behavioural impact of cross-border e-collaboration, and therefore in the intercultural competencies that can be used for e-collaboration between countries with different national cultures. As [24] acknowledged, "it is important to recognize that intercultural competence goes beyond language and beyond knowledge about other cultures" (p. 140). Holmes and O'Neill [25] argued, for example, that communication is embedded within worldviews based upon totally different values, varying from communication in Asian countries that reflects harmony and relationality, in contrast to the more Western-oriented highly verbal communication and task-oriented goals.

Herewith, the central question in this semi-systematic or narrative review is: What is the meaning of ICT in developing countries, specifically, the importance of 'making sense', and can intercultural competencies help us bridge the gap in e-collaboration between countries?

From a managerial point of view, it is important to improve the percentage of successful transfers of technology; after all, different interpretations of technology between societies are often ignored in ICT transfer projects, resulting in supposedly 'perfect' ICT implementation projects, which are dilapidated afterwards [26,27]. Awareness and understanding of the local situation, which is often different from the home country 'logics', may lead to a more sustained and successful implementation of ICT. This implementation process is enhanced by fostering the intercultural competencies that generate this awareness and understanding. Similarly, Liaw [28], in her study about e-learning and intercultural competencies, refers to the importance of developing a user's understanding of key aspects of (Western) work culture and practices.

*Method*

We use a different strategy for conceptualizing the key constructs in our study and for analyzing its outcomes. In our semi-systematic or narrative review approach, we examine ICT adaptation and diffusion in developing countries, wherein these phenomena have been conceptualized differently and are subject to implicit cultural differences. The latter hinders a full systematic review process [29] and, as our aim was to gain more insights into current ICT practices, we investigate how ICT adaptation and diffusion has progressed over time. In general, this semi-systematic review seeks to identify and understand potentially relevant themes and previous and current research traditions that have implications for the optimization of ICT use and collaboration between cultures. The choice of characteristic cases provides us with a real-life basis for a better understanding of the complex phenomena of ICT adaptation and diffusion and is meant to synthesize the insights we distill from these cases using meta-narratives instead of by measuring effect sizes [1,29].

One of the underlying theoretical perspectives that we use comprises Hofstede's cultural framework [30]. We adopt this model as an underlying theoretical perspective for understanding cross-cultural differences and identifying components [31] of, in our case, successful ICT adaptation and diffusion. Specifically, we look for trends and patterns in the world-wide adaptation and diffusion of ICT.

In our literature review about ICT use in developing countries, key themes include ICT adaptation and diffusion, language barriers, cultural attitudes, and economic access and progress. Through a thematic or content analysis building upon Hofstede's model [30], we compare the results and conclusions from six characteristic case studies that emerged from our review.

In the next sections, we will go into the theoretical background of our study and a summary of the case examples dealing with ICT implementation in developing countries. These case studies are illustrative of the cultural determinants in the use of ICT and will be analysed and interpreted following a similar pattern. First, we examine whether the characteristics of Hofstede's model [30] are observable in the cases, and we will focus on the following issues: What questions recur across the cases? And what are debates, conflicts, and contradictions in the adaptation and diffusion of ICT? The resulting insights will help us to understand the importance of intercultural competencies for enhancing the collaboration in developing countries through the use of ICT. Next, we will thoroughly discuss the significance of these findings in relation to the use of different models of intercultural competences, which are used as foundations to overcome the barriers of cultural differences and collaboration. All in all, in our semi-systematic or narrative review, we present four potentially relevant research avenues for analyzing, explaining, and fostering intercultural competencies.

## 2. Theoretical Background

*Cultural Frameworks*

The 21st century is filled with diversity, due to the 'global world' we live in, which is enhanced by numerous ways of technology-driven communication and collaboration, and a variety of different religions, genders, and cultures, with different beliefs, ways of thinking, abilities, and ways of communicating [24]. Culture can be defined as the "collective cognitive programming, which guides the behaviour of people" [32] (p. 25). It includes values, norms, beliefs, and perceptions of people sharing the same culture. Culture is mostly not an item with which people are engaged in on a conscious level. Rather, people 'live in and are a part of a culture', and take many things for granted, as these are 'normal' and part of their culture. Culture comes to the forefront when dealing with people of other cultures — 'they do the things differently than we do'—or when a new artefact is introduced that has not been invented in one's own culture. Implicitly, the cultural values of the 'inventing culture' are 'built into' the artefacts. For example, a car—invented and produced in Europe and the United States—is supposed to transport people from A to B in a fast and mostly comfortable way. But why should one be fast? One answer is that in the 'American way of life', time is money. So, fast mobility is encouraged in that culture. Another answer is that the road infrastructure and legal system allow fast driving. This can be different in developing countries.

Similarly, the fact that an artefact incorporates the values of the 'inventing culture' will also be the case when introducing technology that has been developed in Europe or United States, such as ICT, into developing countries. Then, cultural differences will become more evident, because the 'receiving culture' will use and adapt the new technology (albeit more slowly), in another way, or with other accents. Geertz [33] defined culture as the code or creation of meaning behind a group of human beings' interpretation of life. According to this view, reality is socially constructed, and an outside artefact, such as technology, will in a similar vein go through a social construction process before it is accepted or included in a specific culture. Obviously, it is even possible that the new technology will not be adapted at all.

Schein [34] summarized culture as "a pattern of basic assumptions—invented, discovered, or developed by a given group as it learns to cope with its problems of external adaptation and internal integration—that has worked well enough to be considered valid and, therefore, to be taught to new members as the correct way to perceive, think and feel in relation to those problems" (p. 9). In line with Hofstede [35], this definition builds upon the idea that culture is something relatively stable and deeply rooted in the fundamental values and assumptions of a group about the outside world. From this so-called etic perspective, cultures are perceived as external entities that can be compared, measured, and ranked. Various authors [32,36–38] developed dimensional frameworks that are aimed at describing differences between national cultures. In this paper, we focus on national cultural

values' differences, as these values' differences are embedded at a deeper level in human behavior than corporate cultural values. Furthermore, in the cooperation process between western and developing countries, governments play an important role, next to NGOs, and for both of these types of entities, national cultural values are more relevant. The idea behind this approach is that a culture can be mapped by describing the distinguished dimensions [39,40]. Hofstede's model [30] with its four, later five, and most recently six, dimensions of: (1) power distance; (2) uncertainty avoidance; (3) individualism versus collectivism; (4) masculinity versus femininity; and, later added, (5) long-term versus short-term orientation, and (6) indulgence versus restraint, is most well-known. We consider the Hofstede dimensions [41] appropriate for this study due to its worldwide familiarity and reputation. They are described in Table 1, which will also be used to point out similarities or differences when presenting the cases. For the sake of the interested reader, we also mention the GLOBE study (House et al., 2004) that builds further on the Hofstede values' dimensions and that focuses on culture and leadership. Besides, we notify the values' model by Trompenaars and colleagues (1997), which consists of seven dimensions and is used to develop private business consultancy activities.

**Table 1.** Hofstede's Cultural Dimensions [30].

**Power Distance**

This is the extent to which people accept inequality of power. In a country with high power distance, it is ordinary for the less powerful members of society to accept their position within a hierarchical arrangement without seeking explanations for such inequalities in power. In a low power distance country, on the other hand, people do not passively tolerate inequalities in power; rather, they demand validations for any inequalities in power that may exist and actively seek to eradicate them.

**Individualism versus Collectivism**

This is the extent of self-centrism and closeness of social relations among people of a particular society. In individualistic societies, people are primarily self-centered, being concerned only about the needs of themselves and their immediate family members. Hence, such cultures are characterized by loosely knit social relationships. At the other end of the spectrum, we have closely-knit, collectivist cultures where people identify themselves as part of a wider group. People remain loyal to their group and look after each other in exchange for this loyalty.

**Uncertainty Avoidance**

In strong uncertainty-avoiding cultures, people have difficulty handling unfamiliar, unstructured situations with uncertain outcomes, and they prefer to avoid such situations. Hence, behaviour and codes of beliefs in these countries tend to be inflexible, and people do not tolerate unconventional behaviour and ideas. In contrast, in weak uncertainty-avoiding cultures, people are more open to the view that the future is inherently uncertain and do not adhere to set behaviour and beliefs.

**Masculinity versus Femininity**

This dimension measures the relative importance of certain attributes that are typically associated with masculinity. In masculine cultures, there is an emphasis on ambition, assertiveness, material rewards, and competitiveness. At the other end of the spectrum are feminine cultures, in which people attach greater importance to non-material aspects of life, care for those weaker than them, and are generally tender in nature. Unlike masculine societies that have competition at their core, feminine societies are founded on cooperation and consensus.

**Long-Term versus Short-Term Orientation**

This dimension has its roots in Herman Kahn's hypothesis that Confucian values could have contributed towards the miraculous economic growth enjoyed by some East Asian countries [42]. Long-term orientation refers to a focus on the future, which is displayed through thrift, perseverance, having a sense of shame that dissuades people from doing anything to put one's family into disrepute, and fulfilling mutual obligations imposed by hierarchical social relationships. At the other end of the spectrum is short-term orientation, which is associated with a concentration on the past and present as captured by respecting traditions, saving "face" in front of others, returning favors, and an effort to instill stability into one's life.

**Indulgence versus Restraint**

While indulgence involves the pursuit of happiness by enjoying life and having fun, at the other end of the spectrum we have restraint, which is associated with controlling the gratification of such desires through strongly established social norms. Indulgence results in a tendency for people to remember positive emotions.

## 3. Cultural Differences and ICT Implementation

Research investigating cultural differences concerning the implementation of ICT has been scarce. Earlier work pointed out that culture indeed affects an individual's perception of and choice for communication media [43–47]. For example, Rice, D'Ambra and More [44]

found that respondents from collectivistic countries rated the telephone as less rich, and the business memo as richer, in comparison with respondents from individualistic countries. Lee et al. [48] define type I and type II cultures that impact adoption of technology. Another example, referring to national cultural and institutional differences in this regard, comprises the specific way of educating students in Muslim countries, where an oral tradition goes together with patriarchal teacher–students relationships, based on large power distance. Obviously, this has implications for adapting a new student-centered teaching strategy that goes along with the adoption of ICT in science education in Arab countries [49].

As regards its prevalence, most ICT-use can be found in developed countries while ICT-use is not as widespread in most developing countries. An example comprises the difference in ICT diffusion patterns, being another indicator of adaptation of new technology. For instance, in Japan, which can be classified as a high-context country, cultural proclivity to avoid uncertainty by relying on traditional, information-rich channels for communication, works against the acceptance of asynchronous, lean electronic channels such as E-mail [50].

According to Menguita-Feranil [51], there is also a difference concerning (giving meaning to) the use of ICT between men and women, e.g., women would use ICT relatively less, spend less time using the Internet, and do not have the same levels of access to ICT at work [52–54]. However, a study concerning the use of ICT by refugee women from Myanmar (Burma) compared to refugee women from Thailand, showed what the (positive) impact of ICT can be on their lives. Particularly, the use of ICT (i.e., Internet and mobile phones) had a positive effect on "binding family ties, social networking and expanding relations to a broader perspective as learning arena, venue for advocacy and amplifying women's voices and sites for cultural expression and entertainment" [51] (p. 10). As such, it had a positive influence on their empowerment. Obviously, the ideas behind ICT, as developed in Western cultures, can be questioned when dealing with ICT in other cultures, as became clear when media richness was used to explain media choices in a Confucian virtual workplace in South-Korea. The study by Lee [55] showed that use of e-mail is significantly influenced by a cultural protocol, such as showing respect for seniors, within a Confucian virtual workplace. "When a worker is faced with media choices for communication in a workplace where showing respect is considered essential, information richness of the chosen media is found to be less important than the capability of providing features to convey cultural protocol" [55] (p. 196). Analogously, Miike [56] stated that communication in e.g., East-Asian cultures, is embedded within a culturally defined view that values harmony, relationality, and the important notion of circularity of life and death. In China, there is a preference for silence as a means, even in e-collaboration, to foster harmonious relationships and to maintain face [25,57]. From a Western perspective, e-communication and collaboration are focused merely on verbal task-oriented goals (rationality), which might create tension, as we have just made clear that there are many cultures that can only adapt properly to ICT when they build upon relationality as well.

## 4. Cultural Differences and ICT Adaptation

A specific question with regard to cultural differences that is dealt with in another group of studies pertains to the adaptation process of ICT in developing countries. In this section, we will build upon the argument that culture is the most important factor to be taken into account when it comes to the absorption speed and usage of ICT as a means of collaboration and interaction (see also [58]).

Before we will move on to the cultural implications of ICT adaptation in developing countries, we will first have a look at the technology that is needed for a sound adaptation. The Internet and associated applications are increasingly taking on the classic functions of ICT as a primary tool for efficiency in the workplace. Internet is powered by ICT, but considering the growing user numbers is a very dominant ICT development in itself as well [59]. The worldwide diffusion of Internet is growing rapidly. In that way, ICT encompasses all kinds of technology that facilitate the processing, transfer, and exchange of

information and communication services. In Table 2, we can see the World Internet usage in different regions.

**Table 2.** World Internet usage and population statistics on 30 September 2020 [59].

| World Regions | Population (2020 Est.) | Population % of World | Internet Users | Penetration Rate (% Pop.) | Growth 2000–2019 | Internet Users % |
|---|---|---|---|---|---|---|
| African | 1,340,598,447 | 17.2% | 631,940,772 | 47.1% | 13.90% | 12.8% |
| Asian | 4,294,516,659 | 55.1% | 2,555,636,255 | 59.5% | 2.14% | 51.8% |
| Europe | 834,995,197 | 10.7% | 727,848,547 | 87.2% | 593% | 14.8% |
| Latin America/Caribbean | 654,287,232 | 8.4% | 467,817,332 | 71.5% | 2.49% | 9.5% |
| Middle East | 260,991,690 | 3.3% | 184,856,813 | 70.8% | 5.53% | 3.7% |
| North-America | 368,869,647 | 4.7% | 332,908,868 | 90.3% | 208% | 6.8% |
| Oceania/Australia | 42,690,838 | 0.5% | 28,917,600 | 67.7% | 279% | 0.6% |
| World Total | 7,796,949,710 | 100% | 4,929,926,187 | 63.2% | 1.266% | 100% |

Chau [60] argued that cultural differences influence the diffusion and adaptation of ICT in collaboration settings with developing countries. For example, a difference can be made between high-context cultures (collectivistic cultures, under which lies many developing countries) and low-context cultures (individualistic cultures). In a high-context culture, meanings in communication are not only found in the content, as in a low-context culture, but also in the nature of the situation and the existing relationships. So, collectivistic cultures 'need more face-to-face contact because people depend on context and use non-verbal communication more than individualists do who are quite satisfied with written communications' [61] (190). So, people in collectivist cultures may need more context-rich communication tools. Although the impact of culture on ICT diffusion is present, ref. [62] perceives culture as a double-edged sword: Technological diffusion can be stimulated in one type of culture while in other cultures it may be hindered.

Since the year 2000, governments have been building facilities in the form of infrastructures for reliable transfer and efficient management of information. Besides, many organizations in developed countries have converted to new modern technology in order to gain competitive advantages [63]. Basically, this process refers to adopting, implementing, managing, and integrating information with organizational activities to provide better products and/or services. However, according to Twati [63], adopting any technology, in particular ICT, tends to change the associated work practices, and often necessitates a redesign of the human activity systems in which the specific technology is embedded. Unfortunately, this redesign is often lacking, putting societies and organizations in difficult situations, and sometimes even leading towards failure in ICT adaptation and transfer. That is to say, efforts of technology transfer cannot be meaningful if they do not take account of the local technological and cultural constraints.

Pradhan [18] argued that all developing countries, even neighboring ones, are quite heterogeneous. Although we adhere to the idea of culture-related individual differences between countries that influence specific outcomes of ICT adaptation, the particular studies that are outlined in the next section show obvious similarities and common patterns as well. Let us take a closer look at some classic examples from the early adaptation processes beginning at the end of the last century, by discussing six cases. Most of these examples portray the use of Hofstede's [32] model for research on ICT adaptation.

### 4.1. Indian Government: Expediency

We will start with Madon [64], who described a project of the government of India, where support was given to manage large amounts of collected data by the District Rural Development Agencies. This support comprised facilitating all agencies to purchase personal computers that would offer local-level decision support to the management. This Computerized Rural Information Systems Project (CRISP) was supposed to help increase report generation and to facilitate the availability and analysis of micro-level data for planning at the district level. However, the project was not successful; still, the manual system was used and the CRISP seemed to be largely redundant. Madon [64] identified six cultural factors that were responsible for hindering the use of ICT. First, public administration follows the style of colonial bureaucracies, which tends to be elitist, authoritarian, paternalistic, and inflexible. Second, decision-making in Indian development administration is highly politicized; political expediency replaces rational judgment. Third, a leader in the Indian public administration is like a father figure, which is highly paternalistic. Fourth, caste and status consciousness dominate social relationships between higher- and lower-ranked officers. Fifth, a new computerized system was supposed to make information more visible and transparent, being a risk for eroding a manager's power base within the administration. Sixth, the apparent formalized behaviour in institutions in India is a cover for informal behaviour based on deep-rooted values from a traditional past. It may be clear from these six factors that the transfer to another cultural environment, in this case India, and the implementation of ICT in such an environment will be accompanied by lots of misunderstanding, delay, irritation, and frustration [65].

### 4.2. Libya: Negative Attitudes towards the Internet

In his Libyan case example, Twati [63] claimed that (organizational) culture is the key success variable of Information Systems (IS) adoption, but that effective leadership is the means by which the culture is created and managed. Results of this study about problematic ICT adaptation showed that Libya adheres to a traditional education system with teachers showing authoritarian and paternalistic roles, wherein there is less room for student initiative, which is, however, urgently needed in case of adoption of the use of Internet [66].

Critical issues comprise the influence of older senior management executives in higher education with no information-technical background, the Libyan people's nature of not taking risks or not being in favor of own entrepreneurship, and the low overall education level. Libya is a country wherein the cultural attitude is people-oriented and wherein one focuses only on the end-user of ICT [63]. Moreover, Libya is also characterized by its organizational commitment to members' well-being and loyalty. However, what does this mean for the ICT adaptation process? In order to answer this question, we found some intriguing answers in a study by [50] who did a comparative research on Arab culture dealing with ICT transfer.

In particular, Hill et al. [50] stated that: "as most technology is designed and produced in developed countries, it is culturally biased in favor of the developed countries social and cultural systems" (p. 29). This bias creates cultural and social obstacles for developing countries to transfer technology into practice. We agree with the authors that the existing literature mainly focused on economic factors that have influenced the transfer of technology, although more and more attention is given to the characteristics of national cultures and their interplay with the transfer of technology as well. However, in line with the Libyan case, the question is, to what degree each of the specific characteristics of the specific countries influence the desire to adopt and the success or failure of the adoption, and to what extent differences in these characteristics have an impact on the suitability of existing models and prescriptions on the transfer of technology. More specifically, these questions raised by [50] refer to differences in management styles, religion, and culture, which all act to create complexity for technology-mediated collaboration in the Middle East. All in all, while the number of studies into Arab management styles is limited, those

documented showed considerable differences from the 'expectations and mores' of Western cultures. This will have impacted the adoption of the use of ICT.

In addition to the aforementioned dimensions of Hofstede [35], ref. [50] distinguished between culture-specific dimensions that are intended to measure the cultural differences in the light of successful ICT adaptation. Interestingly, in their study, they found that while under certain conditions the dominant culture in one country tends to put greater emphasis on dimensions like fatalism, conformity, obedience, and charity, as regards ICT use, counter forces in that same country are placing greater significance on free will, creativity, open-mindedness, rebellion, and justice, at the same time. Therefore, in addition to the findings that cultures are not only undergoing changes over time, we posit that a specific culture or a (part of a) society is also a complex system in itself with competing forces.

Another example is where ICT implementation and adaptation can only be initiated as a top-down process. From the study in Arab countries [50], it was obvious that although organizations purchase information systems, many of the top managers did not use it personally. Moreover, in their research, they showed that the hierarchical structure of Arab society dominates the use of ICT, i.e., technological change must come from top management. At the same time, they also witnessed bottom-up movements by which younger people who had studied in the Western world brought ICT and its related cultural values to the organization and into society, yet that hierarchical structure prevented its widespread use. As Twati [50] stated, in Arab society, the knowledge, skills, and new technologies that people learned in other (mostly Western) countries were not to be accepted automatically as viable alternative policies and procedures for existing ones in business and organizations.

Respondents in Twati's [50] study reasoned that, due to a history of colonization, Arabs often show passive behaviour. The authors noticed that, although Arabs talk about changes, they do not always take actions required for change to take place. Consequently, Arabs may say that they support technological innovation while their concrete actions may reject the use of it. More specifically, one of the cultural patterns that is contradictory to the embedded philosophy in Western ICT is that "Arabs share information with one another only if the individual thinks that by doing so, he/she will gain status or power or that their kin people or work group will gain by their actions" (p. 12). Loyalty to religion, family, and national traditions often outweighs accepting change from outside, i.e., brought by ICT. That is to say, a strong uncertainty avoidance, a high power distance, and strong collectivism seem to be relevant here as underlying assumptions with regard to daily behaviour.

### 4.3. Thailand: All about Service

In the research on Thai Tourism Industry, wherein Vatanasakdakul [67] also used the model of Hofstede (1981), the dichotomy of Masculinity and Femininity was found to be relevant. The characteristics of a masculine society are expressed by means of a masculine, assertive, and competitive social role, while a feminine society represents a caring, modest, and weak social role. As a result, in a feminine society, esteem is of less importance than in a masculine culture [68]. For the Thai's feminine society, Vatanasakdakul [67] found that maintaining a warm relationship with people is crucial. Personal relationships including exclusive personal attention, warm interactions, and face-to-face contact are fundamental to building trust. Therefore, providing an on-line service is not viewed in the same meaning from a Thai's point of view, in comparison with the Western society, and therefore hinders the adaptation and use of these technological possibilities.

In Western society, relationships grow out of (business) deals, while in Asia deals are primarily the result of social interactions, involving the exchange of favors [69]. In addition, these relationships are formed and maintained by personal face-to-face relationships. So how can it be replaced by electronic communication?

Furthermore, Hofstede [35,68] asserted that Thailand is a collectivistic society. In collectivistic societies, members are concerned with the group interest rather than with the

individual interest (Ibid.). This introduces a second contradiction, since the ICT adaptation is primarily starting from an individualistic point of view and considers all members of an ICT community to be individuals. Obviously, this does not match with the collectivistic Thai culture. To summarize, the general finding here is that the local Thai culture emphasizes the role of a trusted insider, which is in conflict with Western society's new social network concept that suggests that the new information and opportunities are critical for the survival of any individual [69,70]. In contrast, the development of a 'group app' might be more easily accepted in Thai culture.

*4.4. Ethiopia: Ideology Bends ICT Use*

The case example of Ethiopia deals with the way in which ICT is concerned with the political objectives of the government, i.e., the ruling political party, and wherein ICT is used as a means of exercising power. It is therefore not surprising that the dimension of power distance from [71], a dimension on which Ethiopia scores relatively high, plays an important role in this case. Gagliardone [72] described that, since the 1990s, the government has been formed by the Ethiopian People's Revolutionary Democratic Front. The government is in the process of transforming the state on the basis of two principles, namely ethnic federalism (defined and segregated by ethnicity) and revolutionary democracy. In contrast with a liberal democracy, revolutionary democracy is not focused on the individual. Rather, such a democracy prefers group rights and the emphasis is on consensus. This corresponds to a second important dimension from the work of [71,72], namely collectivism, on which Ethiopia also scores relatively high.

Over the past decades, the government has started the introduction of ICT. Two applications are important here, namely Woredanet and Schoolnet. Woredanet offers a variety of services to local administrations, including Internet connection, e-mail service, and Voice over IP (VoIP) service. Use is made of satellite communication based on Internet Protocol (IP). Woredanet was installed in the remote areas to prevent individuals from independently finding solutions for their problems (i.e., by accessing the Internet) and rather to enable the central authorities to provide clear guidelines when needed. Schoolnet uses a similar architecture to broadcast pre-recorded lessons on a variety of subjects to all secondary schools in Ethiopia. Schoolnet is also used to offer political education to teachers and other government institutions. On the one hand, ICT has the potential to open up societies; on the other hand, as is the case in Ethiopia, there is the ability to encapsulate ICT by an authoritarian government and to restrict its use.

To summarize, the government bends ICT according to their preferences, encapsulated by their ideology, with Woredanet and Schoolnet being two important applications of ICT, and two of the most striking examples in this process. A high power distance and a high collectivism seem responsible for the behavior in the Ethiopian culture.

*4.5. Vietnam: ICT Is 'Ultimately Packed Behavior'*

Another use of Hofstede's [32] model for research on ICT adaptation is found in a study in Vietnam, by [73]. In this country, a basic assumption comprises the belief that the introduction and adoption of new technology improves productivity, and that, in turn, the net social and economic impacts will be positive [73]. What makes this study interesting is the way in which ICT itself is perceived as a form of culture [74]. For instance, William (1974) observed that technologies such as TV, phones, and lighters have actually become cultures in the west, and are no longer thought of as technologies.

Do et al. [73] used the framework of [34] who distinguished between two cultures: A and B. In an A culture, ideas come ultimately from individuals, where truth can only be found by fighting things out in groups, and where people are responsible, motivated, and capable of governing themselves; that is to say, people engage in intense conversations and arguments, and there is a general sphere of informality. In a B culture, on the opposite, truth derives ultimately from older, wiser, and higher-status members, where people are capable of loyalty and discipline in carrying out directions, where relations are basically



linear and vertical, and where people will ward off any invasions. In such a culture, hardly anything is done, except by appointment, and in case people of different ranks are present, there is deference and obedience; that is to say, an air of formality permeates everything in order to reduce uncertainty. Again, we see a high power distance, a high uncertainty avoidance, and a high collectivism, as explanations for the cultural differences (see [32]).

*4.6. Nepal: All about Values*

In our final case example, we present a study from Nepal, where [18] found that ICT came as a package with development projects from various foreign NGOs. These projects primarily focused their activities on the development and delivery of specific outcomes, within specific time frames, due to project planning incorporated in ICT. Little or no provision was made for building the technological capability of the recipient organizations to sustain the use of the new technology beyond the lives of the projects. It was supposed to perpetuate dependency by these organizations on external resources [18] (p. 3). However, implementation of a new technology does not end with installation of the machinery and explanation of how to use it [75]; rather, it should be accompanied by transfers in education, organization, administration, employment, strategy, and research [18] (p. 5). Therefore, the set of values introduced by, and indispensable for the use of, the new technology must not be contradictory to the values of the receiving society [76]. In the case example of Nepal, most practices in work places function on the basis of communalism where individual initiatives are exceptional [77]. The importance of social interaction in the workplace greatly exceeds the importance of solitary technology interaction. There is also no tradition of using written language for internal communication within a family or within a business context. As a result, there is a strong resistance towards communication via ICT [18] (p. 5), and 'the IT development in Nepal is haphazard' [78].

Comparing the cases along the cultural dimensions using Hofstede's model [32] (see Table 3), we found that especially the difference between individualism versus collectivism is characteristic in all examples, followed by power distance, which is the perceived hierarchy and power differentials in the use of ICT, and uncertainty avoidance. The use of ICT appears to be significantly influenced, for example, by the way in which hierarchical stratifications exist in cultures that allow tools, such as e-mail, to be used as a means of communication in the hierarchy, or by the extent to which ICT is allowed to gain access to certain sources of information, or by the extent to which ICT use is permitted at all. Regarding the dimension of individualism, the question is whether in collectivistic cultures a more individually-oriented ICT application is appreciated or understood. It also seems that in order to prevent uncertainty, one adopts an increasing formalization. Also check in Table 3 the large differences in some of these dimensions between on one hand the countries from the case studies e.g., India, Libya, Thailand, Ethiopia, Vietnam, and Nepal, and on the other hand Western countries such as the United States and the Netherlands.

**Table 3.** Summary of observing Hofstede's dimensions [32] in our sample.

| Hofstede's Dimensions | India | Libya | Thailand | Ethiopia | Vietnam | Nepal | US | The Netherlands |
|---|---|---|---|---|---|---|---|---|
| Power distance | 77 | 80 | 64 | 70 | 70 | 60 | 40 | 38 |
| Individualism versus collectivism | 48 | 38 | 20 | 20 | 20 | 30 | 91 | 80 |
| Uncertainty avoidance | 40 | 68 | 64 | 55 | | 40 | 46 | 53 |
| Masculinity versus femininity | 56 | 52 | 34 | 65 | 40 | 40 | 62 | 14 |

**Table 3.** *Cont.*

| Hofstede's Dimensions | India | Libya | Thailand | Ethiopia | Vietnam | Nepal | US | The Netherlands |
|---|---|---|---|---|---|---|---|---|
| Long-term versus short-term orientation | 51 | 23 | 32 | Not available | 57 | Close to China and India, estimation: 70 | 26 | 67 |
| Indulgence versus restraint | 26 | 34 | 45 | X6 | 35 | Close to India: estimation: 25 | 68 | 68 |

Now that we have discussed these six case examples, in the next section, we will explore possible ways for bridging the cultural gaps during e-collaboration through intercultural competences.

## 5. Intercultural Competencies for Bridging the Gap

Intercultural competencies can be defined as "a complex of abilities that are needed to interact with people from other cultures adequately and effectively" [79] (p. 488). Developing intercultural competencies is a solution for understanding and possibly coping with the specific cultural aspects troubling the use of ICT in developing countries. More specifically, the main issues besides ICT adaptation in different cultures comprise: exchanging knowledge, learning, and e-collaboration, in our case between governments and NGOs, and the rapid growth of ICT as a highly important way of communication. So far, we have mainly looked at the process of implementation of ICT in developing countries. A preliminary conclusion from this endeavor is that the success of ICT implementation is highly dependent on the underlying cultural characteristics that encompass Western ICT technology versus the meaning given to it by other cultures. That is to say, ICT is culturally-biased by Western culture that does not always reflect the values that other cultures experience, like, for instance, in the use of e-mail as a means of communication. As a result, and like [80] argued: "people from different countries see, interpret, and evaluate events differently, and consequently act upon them differently" (p. 72).

While the world gets smaller ('global village') and the amount of international communication and collaboration increases rapidly, more cultural differences surface, and herewith the call for intercultural understanding is greater than ever [81]. That is to say, we doubt whether it is possible from our own Western point of view to investigate the cultural impact of ICT in developing countries without having a tool that accounts for cultural differences. In the next section, we will continue supporting our view in this regard. As Simpson and Dervin [82] (p. 673) already stated, the notion of intercultural competencies itself can be regarded as a "victim of Western-centrisms", meaning it contains "politicized discourses and practices, but also idealistic 'postmodern' ideology".

Intercultural competencies may (partly) fill the gap regarding human capacity that is needed to foster the use of ICT in projects wherein developed and developing countries collaborate [83]. Yet, what do we know about these competencies? Over recent years, several studies have shown the importance of an intercultural competence framework for understanding cultural differences in effective communication [6,84–89]. These models include the ability of understanding and being aware of sociolinguistic, specific patterns, roles, and impact of one's own culture and that of partners in communication and collaboration. Spitzberg and Changnon [90] argued that intercultural competence, however, is largely viewed as "an individual and trait concept" (p. 44) that does not take into account the psychological and emotional aspects of communication [25].

It is also argued that intercultural communication is just a special case of interpersonal communication and not a unique field of study and practice [91]. Let us explore some of the theories and models that are used for intercultural competence in more detail.



However, before we will do so, we will first further elaborate on the concept of intercultural competencies.

Although there have been many attempts to understand possible differences across cultures, one can argue that we still do not have a full picture yet (e.g., [92]). The simple fact that we see ICT as a universally understood way of communicating over the world comprises an important example. There is hardly any questioning about the culturally-biased way ICT is built and used, starting from, for instance, the limited Western view of how the world works. However, it seems necessary that we need to agree, at least, upon a form of common ground, that is to say a kind of neutral base.

ICT is strongly built upon statements about facts, values, and action principles, and therefore incorporates a way of ethnocentrism. In the global village (where the impact of Internet and ICT brings people 'closer' to each other, by means of modern ways of communication), we all interact with people who have different values, behavioral norms, and ways of perceiving reality. Therefore, the need to develop intercultural competencies has become more and more important [87,93], and refers to the need to understand, reflect, and act on the culturally shaped expectations and norms of our counterparts in each new situation.

The kind of intercultural competencies required today is the ability to recognize and to use cultural differences as a resource for learning, and for the design of effective action in specific contexts [84]. Berthoin Antal and Friedman [84] assumed that "the more people differ, the more they have to teach and learn from each other" (p. 1). "To do so, of course, there must be mutual respect and sufficient curiosity to overcome the frustrations about these misunderstandings" [94] (p. 51). This critical cultural awareness emerges through self-reflection at the time of the interaction [95]. Holmes and O'Neill [25] argued that developing intercultural competencies involves making judgments about one's own communication in the interaction and then reflecting on and learning from those judgments. Building on Holmes and O'Neill [25], we will discuss four models of intercultural competencies that differ in clarifying differences and similarities between cultural determinants, and that may be of help in bridging the cultural gap, and in dealing with the (in)congruencies, complementarities, and dissonances within and between countries

### 5.1. Intercultural Competencies: The Six Stages' Model by Bennet

Bennett [96] developed a commonly used model differentiating between six stages of awareness regarding cultural differences. These six stages represent: (1) denial of difference; (2) defense; (3) minimization; (4) acceptance; (5) adaptation; and (6) integration, being a final stage where people are capable of reconciling cultural differences and of forging a multicultural identity [96]. During the first three stages (denial, defense, and minimization), participants in communication fear that these differences threaten them, or they seek to minimize differences and, as a result, ignore difficulties and misunderstandings in intercultural communication. The result is that they are unlikely to resolve problems or to learn from it. The underlying stance that is common to the first three stages is ethnocentrism, that is, the sense that one's own culture is better than others [97]. The fourth stage, acceptance, deals with this problem of ethnocentrism and provides space for the recognition that other cultures have equally valid ways of seeing and doing things. However, it does not solve the problem of misunderstandings yet. The risk of paralysis can occur, because accepting differences in an oversimplified way is not sufficient when joint decisions and follow-up are required. Rather, steps five and six, that is, adaptation and integration, are also necessary to fully learn from the cultural differences.

Berthoin Antal and Friedman [84], who criticized the so-called *six stages' model* by [96], referred to it as being quite deterministic, implying that the behaviour to which one should adapt is relatively predictable as in 'touristic' culture guides. The model assumes that if people know enough about different cultures they can "intentionally shift into a different cultural frame of reference" [98] (p. 28) and modify their behaviour to fit the norms of another culture. The recommendations found in such 'touristic' culture guides may suffice

for adapting to the greeting and eating rituals of a foreign culture, and they may provide guidance for adjusting to the business meeting practices in various countries. However, they are inadequate for dealing with the dynamics of interactions between people, who want to make and implement decisions in different contexts (see also [99]). Rather, " ... , what is called for, is an approach that involves considerable observation and listening, experimentation and risk taking—and above all, active involvement with others" [100] (p. 141). Therefore, in the next section, we will outline an alternative approach that regards intercultural competencies as negotiating reality, by providing an actionable means to bringing the skills of observing, listening, and experimenting along with those of reflecting, expressing, and inquiring in order to make intercultural interactions effective and enabling them to generate a richer repertoire for action strategies in the future [84].

### 5.2. Intercultural Competencies: The Model of Advocacy with Inquiry by Berthoin Antal & Friedman

Berthoin Antal and Friedman [84] have coined the term "negotiating reality" to name the process whereby individuals generate an effective strategy of action during an intercultural interaction, by making themselves and each other aware of their culturally-shaped interpretations and responses to a given situation, and by expanding their repertoire appropriately. Negotiating reality involves having the ability to surface the tacit knowledge and assumptions belonging to the parties involved, and to bring this knowledge in service of addressing a particular issue or problematic situation. Negotiating reality is an approach for generating the necessary cultural knowledge for a situation as it arises, in a conversation, and with this knowledge, constructing an effective action strategy. It is a strategy that is less demanding than gathering information, as [88] emphasized in his article. It means that intercultural communication is not about storing as much knowledge about as many other cultures in preparation for the eventuality of meeting with people from other cultures. Yet, it is about learning to use this knowledge in action.

Negotiating reality requires personal mastery, because individuals must have an active awareness of how their own cultural backgrounds influence their perceptions and behaviour, the ability to engage with others to explore assumptions, and, furthermore, the amount of openness to try out different ways of seeing and doing things. Three beliefs underlie the concept of negotiating reality: (1) As human beings, all people are of equal importance and worthy of equal respect; (2) As cultural beings, people differ because they possess different repertoires of ways of seeing and doing things; and (3) The repertoire of no individual or group merits a priori superiority or right to dominance [84].

Correspondingly, Berthoin Antal and Friedman [84] argued that the process of negotiating reality, fundamentally, involves the various parties asking three questions: (1) How they perceive the situation; (2) What they wish to achieve in that situation; and (3) Which action strategy they intend to employ to achieve their goals. By opening themselves to answering one or all of these three questions, the participating individuals become capable of significant 'double-loop' learning [101]. Double-loop learning delivers the groundwork for expanding the repertoires of potential responses to situations involving different culturally complex beings [78,84]. The primary focus therefore should be 'actionable knowledge' as means to this 'double loop learning'. According to Argyris [102], if you act in 'such a way', the following will likely occur ... (instrumental preposition), and therefore will produce (simplified) 'if-then-'propositions that can be stored in and retrieved from the actor's mind under conditions of everyday life. However, what happens if the context changes, and the relevance of previously held actionable knowledge, which is culturally determined by ways of viewing reality and solving problems, becomes questionable [84]? Intercultural situations, therefore, require the development of new actionable knowledge.

Berthoin Antal and Friedman [84] developed the so-called model of advocacy with inquiry as a means of exploring and testing theories of action and reality images collaboratively with the other person or with people involved in the intercultural interaction. Advocacy means clearly expressing and standing up for what one thinks and desires. Inquiry means exploring and questioning both one's own reasoning and the reasoning of

others. It often requires a conscious effort to suspend judgment, to experience doubt, and to accept a degree of uncertainty until a new understanding is achieved [103,104]. Although the model of advocacy with inquiry is useful for meeting the requirements for bridging intercultural differences, it does not take into account the process that leads to the desired action. Therefore, in the following section, we will describe this particular process.

*5.3. Intercultural Competencies: The Pyramid Model of Intercultural Competence by Deardorff (2004–2015)*

A more process-oriented approach can be found in the research by [85]. Based on the data generated from intercultural experts, Deardorff defined cultural competencies as: "the ability to communicate effectively and appropriately in intercultural situations based on one's intercultural knowledge, skills, and attitudes" (p. 184). One surprising result of Deardorff's study comprised the specific skills that emerged through consensus between the participants in intercultural communication, incorporating their skills to analyze, interpret, and relate, as well as skills to listen and observe. The cognitive skills that emerged from this study include comparative thinking skills and cognitive flexibility. These skills point to the importance of processing in acquiring intercultural competencies, and the attention that needs to be paid to developing these critical skills.

Let us take a closer look at the [85] model. The visual representation of this Process Model of Intercultural Competence, originally developed by [85] (see Figure 2), forms a more accessible representation of long-fragmented lists, by placing components of intercultural competence within a visual framework that can be entered through various levels of the model. Attitude is a fundamental starting-point [105] as illustrated in this visual representation. It has been referred to as the affective filter in other models [106], although this filtering process differs from one person to another. Moreover, attitude will vary more by what we bring to the learning than by what we have learned. The model of [105,107] concurs with the ones from other scholars in emphasizing the importance of attitude to the learning that follows. Specifically, the attitudes of openness, respect (valuing all cultures), and curiosity and discovery (tolerating ambiguity) are viewed as fundamental to intercultural competence.

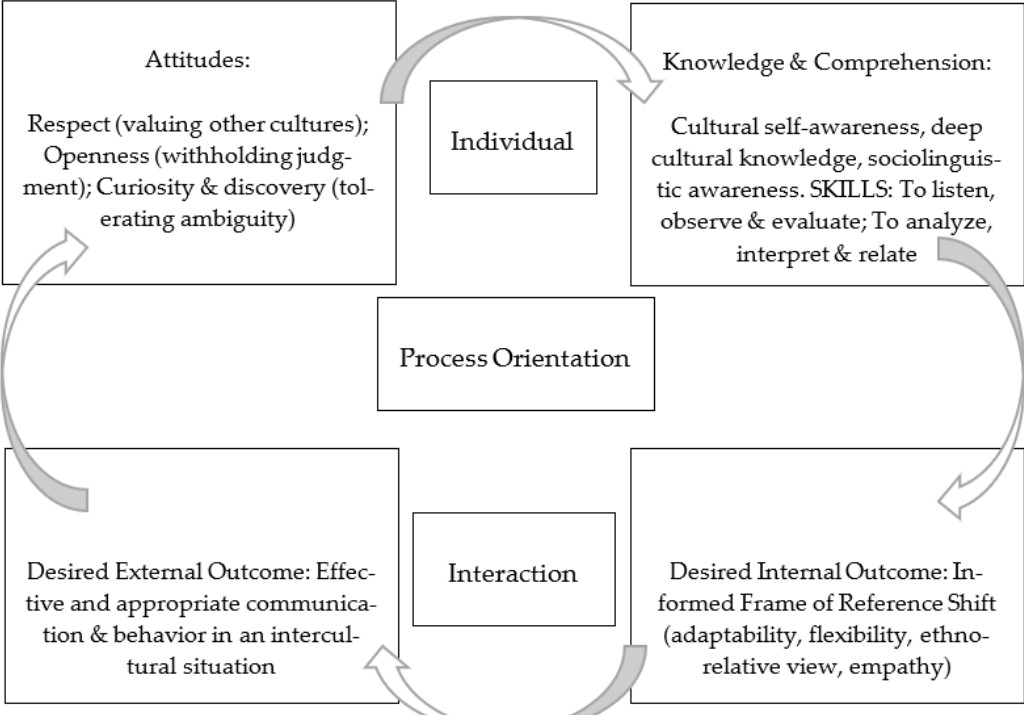

**Figure 2.** Process Model of Intercultural Competence [105].

This model of intercultural competence allows for a differentiation according to the degree of competence (the more components acquired/developed increases the probability of a higher degree of intercultural competence). Deardorff's [85] model enables the development of specific assessment indicators within a context/situation while also providing a basis for general assessment of intercultural competence, thus embracing both general and specific definitions of intercultural competence. The model moves from the individual level of attitudes/personal attributes to the interactive cultural level with regard to the outcomes. The specific skills that are delineated in this model are skills for acquiring and processing knowledge about other cultures as well as one's own culture. Next to attitude, the model also emphasizes the importance of comprehension of knowledge [105] and therefore can be regarded as an integration of the notions of the described models in the previous section.

Deardorff [105] emphasizes the internal and external outcomes of intercultural competence. The internal outcome, which involves an internal shift in frame of reference, while not requisite, enhances the external (observable) outcome of intercultural competence. The external outcome can be described as "Behaving and communicating appropriately and effectively in intercultural situations" (p. 196). Definitions of effective and appropriate are taken from [108] work where appropriateness is the avoidance of violating valued rules, and effectiveness is the achievement of valued objectives.

In a more recent study, Deardorff [107] (p. 142) emphasized the skills of her model as an assembly of observing, listening, evaluating, analyzing, interpreting, and relating. She characterized this assembly as the importance of 'intercultural thinking' in which critical self-reflection is essential for the development and assessment of intercultural competence. Moreover, she referred to her revised model of intercultural competence development as a continuous and cyclic process for acquiring the necessary critical thinking capacity and named it the pyramid model of intercultural competence. Attitudes—especially respect, openness and curiosity—serve as the basis for this model and influence all other aspects of intercultural competence [105]. It is also important to embed intercultural perspectives on the actual subject into e-collaboration and communication, in order to improve the ability to understand people with other worldviews.

*5.4. Intercultural Competencies: The PEER Model of Holmes & O'Neill*

In their PEER model, Holmes and O'Neill [25] emphasized the attention that must be paid to the physiological and emotional aspects of communication in intercultural relations. From this perspective, they defined intercultural competencies as "the effective management of the interaction between people who have different or divergent affective, cognitive and behavioral orientations towards the world" (p. 7). The authors pointed to the effect of Eurocentrism (Western-oriented research institutes and the predominant design of an ample amount of research) in defining and researching intercultural competencies. The focus on management of the interaction also insufficiently recognizes the understanding of the processes that individuals undergo in developing their intercultural competencies.

The authors think that research into intercultural competencies is most clearly visible within intercultural encounters. They use the so-called PEER model of Preparing, Engaging, Evaluating, and Reflecting upon competencies within intercultural encounters. Their findings indicated that "developing intercultural competence encompasses processes of acknowledging reluctance and fear, foregrounding and questioning stereotypes, monitoring feelings and emotions, working through confusion, and grappling with complexity" (p. 707).

The criticism on many intercultural competence models indicates the limited attention for relationships in intercultural encounters, particularly due to Western-oriented research styles and propositions [108]. As an example, Zaharna [109] and Nwosu [110] concluded from their studies that the meaning and purpose of communication derive from relationships between parties, and that people perceive and construct their identity through their relationships with others. The PEER model emphasizes the main questions of: 'Who am 'I' in these interactions? And who am I in the eyes of my cultural 'other'?

The development of intercultural competencies requires judgment about one's own communication with those of the other, and then reflection on and learning from those judgments. This creates an understanding of mutual (in)congruences, complementarities and dissonances [25]. The authors recognize the need for bridging of what they call 'feelings of discomfort' and at the same time 'discovery': The participants in the intercultural encounter have to perceive their own view, as well as the worldview of the other, and their attitudes towards family and political/historical constructions of national religious and other identities. In this way, they become aware of their routine and culture-specific behaviors that manifest themselves and that influence the intercultural encounters. What can be seen as appropriate intercultural communication is created by examining, what the authors call 'typifications' (classifications and patterns of social experience) and 'being social' (the strategies to learn to understand the ways of saying and doing things in certain situations). This also provides the basis for the necessary self-confidence about the question of who someone is, and the appreciation for their own unique culture [25].

## 6. Discussion

This contribution was aimed to come up with a critical reflection upon how the use of ICT is influenced by cultural determinants. Many ICT systems are embedded in the culture of the programmers and, as a result, each culture uses ICT differently, corresponding to its own values and norms. However, what happens when these systems are created in one culture and used in another? And what does this mean for cross-cultural collaboration when making use of these ICT applications? This is most acute in developing countries where ICT is seen from Western ideas as a possible "engine of progress". We have started with an outline of the possible effects of cultural differences in the light of both the implementation and the following adaptation of ICT. Subsequently, six case examples from across the globe have been reported in order to illustrate which classic problems may be encountered concerning the use of ICT across culturally different situation. Following our main conclusion, referring to the importance of intercultural competencies, four models have been described that may help us to come up with clear guidelines to bridge the gap between cultures. We posit that increased insight into necessary intercultural competencies may enable us to more successfully implement and adopt ICT and to facilitate possible collaboration across the globe.

In fact, the different intercultural competencies' frameworks all end up similar to the basic model of [111] in which intercultural competencies are hypothesized to derive from motivation, knowledge, and skills. In general, the different frameworks consist of elements that are described as the ability to negotiate reality, actionable knowledge, sensitivity, engaging strategies, and learning capabilities.

At the start of the scholarly project that is reported in this article, the main concern was that ICT is strongly built upon statements about culturally-based facts, values, and action principles and, therefore, entails a way of ethnocentrism, which is not understood across different cultures. The latter may lead to an obstruction of the successful interpretation of underlying goals and desired outcomes of communication and collaboration wherein ICT is used as an important tool. After having studied some case examples of ICT adaptation in developing countries, the overall conclusion is that the absence of attention to the underlying values of human behaviour in cross-cultural ICT use is one of the most important constraints in effective ICT transfer. These constraints are a barrier for the ability to understand and be aware of sociolinguistic influences, specific behavioral patterns, roles, and to grasp the impact of one's own culture and that of partners in communication and collaboration.

On the other hand, previous research shows different approaches to the use and implementation of so-called intercultural communication skills, or of competencies that can be a solution to the specified problems. Hammer and Bennett [98] captured the acquiring process of such competencies into their more static six stages' model, in contrast to [107,108] who incorporated the personal dimension of relational competencies that foster

the communication by motivation, knowledge and skills, herewith implying a view that already does justice to the notion that in order to enhance one's intercultural communication people need to invest time and energy (i.e., a more dynamic view instead of the static one) in developing these competencies [112]. Indeed, Berthoin Antal and Friedman [84] argued that the kinds of intercultural competencies that are required today form the abilities to recognize and use cultural differences as a resource for learning, and for the design of effective action in specific contexts. Intercultural competencies, therefore, are created by acquiring knowledge of a culture and by the ability to actively use that knowledge in intercultural communication. These competencies enable people to discover different views of reality, making it more likely that they will create common understandings in order to generate collaborative action. This whole process of cultural communication can be seen as "negotiating reality" [84].

Subsequently, Deardorff [85], in her pyramid model of intercultural competencies, referred to specific skills for acquiring and processing knowledge about other cultures as well as one's own culture. For that purpose, ref. [5] proposed the use of e-collaboration systems that enable access to information on cross-cultural peculiarities, encouraging open-mindedness through discussing the information, creating tools of interpreting and reacting to cross-cultural contexts, and modeling the situations to implement cross-cultural knowledge [113].

Finally, Holmes and O'Neill [25], in their PEER model, added relationality to the existing models, herewith clarifying the meaning and purpose of communication that derives from relations between parties, and enabling people to perceive and construct their identity through their relationships with others (role of egality and power differences).

Altogether, the different models complement each other, certainly with regard to attempts to acquire and apply intercultural competencies. In our view, Deardorff's pyramid [85] and later process model [105] offers the most practical guidance for adopting fruitful attitudes towards technology-mediated collaboration.

To more clearly explain the added value of using Deardorff's model [105], incorporating some knowledge gained from Hofstede's 6-dimensional model, we explore how we can adopt it in the first case about IT use in India:

In this specific case, a new initiative of supporting local employees of the government by providing them with their own personal computers was launched in order to give employees at a local level more insight into relevant data and its interpretation. Although meant to enable them to make better and faster decisions, the approach failed due to the fact that these local employees did not feel legitimized to have access to these data, to interpret them, and to take decisions. From Hofstede's perspective [32], we recognize the dimensions of power distance, uncertainty avoidance, and collectivism (that appear to be different in comparing e.g., India and the US). The Indian employee or lower manager will not autonomously start working on using his/her personal computer as they will wait for permission from their boss. Furthermore, as a result of uncertainty avoidance, the employee or lower manager will also wait for precise instructions provided by the producer or as available in the computer programs themselves, telling how to correctly use one's personal computer. Furthermore, the personal computer belongs to the department thatthe employee or manager is employed to; so the department's head would first have to decide how to work with it, referring to the strong collectivistic attitude in Indian work places. These personal, culturally-based perceptions of the employees and lower managers, in this particular setting, are different from the intended use of ICT by its developers. The latter can lead to inactivity and a reactive instead of a proactive attitude. Building on the process model of Deardorff [105], we argue that, before one starts introducing personal computers for all local governments, both the attitudes and the knowledge of the local employees must be carefully outlined. More specifically, in order to support cultural competence development amongst local employees, first, employees' attitudes need to be explored in-depth, with the atmosphere of an open climate, a climate wherein curiosity and learning are promoted, and wherein there is respect for each other needs. Only under

these circumstances, the development of cultural competencies can be fostered. Obviously, creating such a climate and stimulating cultural competence development needs time and sound coaching expertise.

To sum up, using Hofstede's model [32] across the different cases that have been dealt with in our study shows that there are significant cultural differences between countries about the perception and use of ICT and its applications. The latter are rationally, efficiently programmed, especially for individual or group use in the US and Europe, and characterized by a Western perception of work. In short, they are developed in countries with considerably different scores on the Hofstede's dimensions [32] than in the countries where they are to be implemented as well. Using Deardorff's process model [105] in particular, we propose that it is of utmost importance to develop an attitude at work that recognizes cultural differences (based upon openness, curiosity, and respect), thus promoting cultural self-awareness. A sound dialogue in mutual cooperation is aimed at achieving a reference shift that, on the one hand, brings to the fore the possibilities and limitations of the programmed ICT tools and, on the other hand, is precisely the vehicle to enable cultural recognition and cooperation. We regard intercultural competencies as conditional for successful use and adoption of ICT collaboration between cultures As such, it is much more than a user training course or a good manual. A thorough development trajectory should result in, as Deardorff [105] claims, internal and external outcomes that are effective and appropriate communication & behavior in an intercultural situation using ICT.

Furthermore, Deardorff's model [105,106] comprises a model seeing intercultural competencies as a process of learning from acquiring knowledge to, ultimately, also using this knowledge and reflecting on it. This makes Deardorff's process model [105], which we would like to label as a model of action, much more dynamic and helpful to understand the process of learning and knowledge transfer.

Another important insight of our study points to the importance of self-reflection. Participants in e-collaboration are encouraged to look back on their own culture and to grow to a deeper understanding of both the other and their own culture. Hanafizadeh et al. [5] came to the conclusion that these different cultures ultimately show more similarities than differences when it comes to solving a problem.

## 7. Future Research

This study has limitations. We presented six cases to illustrate the culturally different interpretation and use of ICT, which may also lead to several risks of miscommunication and conflicts in intercultural collaboration, e.g., between governments and NGOs. As such, this study contributes to awareness and understanding of the less bright sides of the role of ICT and Internet in worldwide collaboration. The cultural model we have used is a classic model that has proven its value but that also has received some more criticism over the last decade. Furthermore, we distinguish between developing and developed countries, terms that were very popular over the last four decades; however, due to the strong development of emerging markets, the differences between developed and developing countries has become more blurred. All in all, we see the emergence of a new world order with a stronger role for Asian countries such as China and India, South-American economies such as Brazil, and African countries such as South Africa and Nigeria.

Obviously, further empirical research using the specific intercultural competencies' approaches should be performed to more safely conclude about the predictive validity of specific competencies in the light of a successful implementation and adaptation of ICT across countries. In this study, we have used a limited collection of information as we have not covered all articles ever published on the topic. Our intention was to combine different theoretical perspectives in order to create a foundation for a structured, more integrative review approach and for future scholarly work in this field.

It is, for instance, highly interesting to incorporate cultural values' factors (such as the ones distinguished in the Hofstede framework [32]) as possible moderators in future

empirical work. Obviously, more scholarly work is also needed to better understand the possible impact of multiple demographic variables, and other sources of dissimilarity between people working together in project teams, such as personal values, attitudes, and personality (see also [114,115].

Knowledge on the advantages of ICT in relation to a certain country and its context, could be particularly advantageous when planning future ICT and development projects [112]. More research focusing on technology-mediated collaborations is therefore needed to investigate whether the impact of relational norms may differ according to national culture or racio-ethnicity, gender, and age (see [116] for an interesting example in this regard). For instance, the relational norm of age could be more salient in cultures that value and respect the wisdom of the elderly. We believe that the importance of intercultural competencies' models in the light of technology-mediated collaborations are noteworthy and provide opportunities for future research on cross-border knowledge transfer, and cross-validation of the outlined models in different occupational settings and countries [117].

Ideally, this research adopts a longitudinal approach in order to determine causal relationships between antecedents of technology-mediated collaboration, such as intercultural competencies, and its outcomes, for instance in terms of firm and societal success. It is also important to investigate whether intra-individual changes in intercultural competencies across life stages, for instances as a result of building up these competencies through more experiences, and or training and development, throughout one's career, are related to one's career success. We argue that intercultural competencies are highly important resources in nowadays' labor market, which is characterized by an increasing internationalization [118].

To conclude, especially in the beginning of this scholarly project, we focused on the differences between on the one side developed countries, that are in many circumstances responsible for delivering ICT systems that are mostly developed from a Western perspective, and developing countries that make use of these ICT applications and that work from their culturally-formed perspective using these applications. However, obviously the need for making use of intercultural competencies can come from all countries across the globe. It is not self-evident that in collaborations between the powerful West and the less powerful non-Western countries both parties will show initiative to develop cultural competencies and to use them not in order to better interpret the ICT applications developed from the other culture's perspective. For instance, in countries wherein a higher level of collectivism, power distance, and uncertainty avoidance are ingrained in daily life, different systemic approaches are needed in order to develop ICT infrastructure and software [113]. More research is needed in order to disentangle what determinants influence the success and failure rates of e-collaboration between countries across the globe. We hope that our contribution forms a sound basis for designing such research and that its outcomes will increase our insights in fruitful avenues for fostering technology-mediated collaboration with developing countries.

**Author Contributions:** Conceptualization, A.K., R.O., B.V.d.H., and J.B.; methodology, A.K. and R.O.; validation, A.K., R.O., B.V.d.H., and J.B.; formal analysis, A.K. and R.O.; investigation, A.K., R.O., B.V.d.H., and J.B.; writing—original draft preparation, A.K., R.O., B.V.d.H. and J.B. writing—review and editing, A.K., R.O., B.V.d.H., and J.B. All authors have read and agreed to the published version of the manuscript.

**Funding:** This research received no external funding.

**Institutional Review Board Statement:** Not applicable.

**Informed Consent Statement:** Not applicable.

**Data Availability Statement:** Not applicable.

**Conflicts of Interest:** The authors declare no conflict of interest.

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
