# Peer review of "Intercultural Competencies for Fostering Technology-Mediated Collaboration in Developing Countries"

_sustainability, doi:10.3390/su13147790_

Round 1
Reviewer 1 Report
This paper examines intercultural competencies in terms of technology-Mediated collaboration using ICT for 5 cases. The study employs various frame works to pursue what authors want. Throughout 5 cases, authors have tried to find why each country has differ adaptation and diffusion of ICT and which factors and hinder or facilitate the collaboration with developed counties.
- First, it is necessary to provide a correlation to the direct impact of the use of ICT on the spread of culture
- A reviewer thought that this paper’s research purpose is unclear. Thus, there is ambiguity in the background of the research on the relationship between cross-cultural competency cultivation and the use of ICT technology. As authors already know, culture is being composed of various dimensions. For this reason, it seems necessary to reduce the scope of the research topic. It is necessary to elaborate that what cultural element that can be measured by ICT is rather than broad approach like knowledge, information.
- A reviewer think that it would be to measure the countries in five cases with the same cultural criteria.
The number of pages in this paper seems to be large. If you accept it according to my request, I expect the thesis page to be reduced. I’d like to thank you very much for the opportunity to read a good paper. Thanks to the researchers
Author Response
This paper examines intercultural competencies in terms of technology-Mediated collaboration using ICT for 5 cases. The study employs various frame works to pursue what authors want. Throughout 5 cases, authors have tried to find why each country has differ adaptation and diffusion of ICT and which factors and hinder or facilitate the collaboration with developed counties.
Dear Reviewer, thank you very much for your time and efforts, and for the constructive feedback that has helped us a lot to improve our manuscript.
First, it is necessary to provide a correlation to the direct impact of the use of ICT on the spread of culture.
Dear Reviewer, thank you very much for your constructive feedback. In the revised manuscript, we have now established more clarity on the impact of the use of ICT on the spread of culture. See the following text excerpt on page 2 and 3 (lines 94-96): “as there is concern about Western cultural hegemony regarding ‘the information society’ over other world regions [101]; in the central question on page 4 (lines 157-159): “What is the meaning of ICT in developing countries, specifically, the importance of ‘making sense’, and can intercultural competencies help us to bridge the gap in e-collaboration between countries?”, and on lines 174-183: “We will first discuss Hofstede's cultural framework [102] as a basic model for understanding cross-cultural differences after which we will look specifically at the relationship between cultural differences and ICT”.
A reviewer thought that this paper’s research purpose is unclear. Thus, there is ambiguity in the background of the research on the relationship between cross-cultural competency cultivation and the use of ICT technology. As authors already know, culture is being composed of various dimensions. For this reason, it seems necessary to reduce the scope of the research topic.
Dear Reviewer, we have now reduced the scope of the research topic and have reformulated the purpose and research questions in the Introduction section. Please, see the following tekst excerpt on Page 3 (lines 131-133): “Basically, the main concern in this article is an exploration of the meaning of ICT in developing countries ..”and Page 4 (lines 157-159): “What is the meaning of ICT in developing countries, specifically, the importance of ‘making sense’, and can intercultural competencies help us to bridge the gap in e-collaboration between countries? And on lines 174-176: “We will first discuss Hofstede's cultural framework [102] as a basic model for understanding cross-cultural differences after which we will look specifically at the relationship between cultural differences and ICT”.
A reviewer think that it would be to measure the countries in five cases with the same cultural criteria.
Dear Reviewer, thank you for this remark. In order to be consistent, we have now stressed throughout the document that we build our categorization of countries around the framework by Hofstede. See the following text excerpt on Page 12 (lines 517-528): “Comparing the cases along the cultural dimensions using Hofstede’s model, we found that especially the difference between individualism versus collectivism is characteristic in all examples, followed by power distance, that is the perceived hierarchy and power differentials in the use of ICT, and uncertainty avoidance. The use of ICT appears to be significantly influenced, for example, by the way in which hierarchical stratifications exist in cultures that allow tools, such as e-mail, to be used as a means of communication in the hierarchy, or by the extent to which ICT is allowed to gain access to certain sources of information, or by the extent to which ICT use is permitted at all. Regarding the dimension of individualism, the question is whether in collectivistic cultures a more individually-oriented ICT application is appreciated or understood. It also seems that in order to prevent uncertainty, one adopts an increasing formalization” We furthermore added a table to summarize these findings (table 3, page 13, line 532)
The number of pages in this paper seems to be large. If you accept it according to my request, I expect the thesis page to be reduced. I’d like to thank you very much for the opportunity to read a good paper. Thanks to the researchers.
Dear Reviewer, thank you again for your conscientiousness and constructive feedback. The total amount of pages is now 22.
Reviewer 2 Report
I think in this paper there are some biased generalisations and pre-assumptions about the cultures of western/non-western which are not properly backed by significant previous empirical research. Not necessarily poor is connected to non-western. Such claims must be neutralised (or show by empirical evidence)
Comparisons of the countries in section 4 are arbitrary and based on a non-systematic literature study, yet this is the main section backing up the claims of difference. But there is no control set to compare with, resulting in diluting the strength of the conclusion.
The theoretical reference of Hofstede indicators is not clear. These indicators differ a lot even for European countries. So how can it be concluded that the low diffusion of ICT in developing countries and differences of Hofstede?
I think the main problem in this manuscript is the mix of methods and the concepts discussed. the solutions and problems are discussed in random order. I think reorganising the problems and solutions (methods) separately will improve the paper a lot. Clear research questions are needed, since the paper discusses many dimensions, such as rich/poor, communication, developed/developing regions, culture, gender, etc.. which it is extremely had to systematically look into all these factors and a high level as global in a single paper.
And I see a lack of scientific rigour in the conducting of the method. If they could convert to a Systematic Literature study it might save the purpose.
Author Response
I think in this paper there are some biased generalizations and pre-assumptions about the cultures of Western/Non-Western which are not properly backed by significant previous empirical research. Not necessarily poor is connected to non-western. Such claims must be neutralised (or show by empirical evidence).
Dear Reviewer, thank you very much for your time and efforts, and for the constructive feedback that has helped us a lot to improve our manuscript.
Although we have tried our utmost to be neutral in our formulations, we are with you that the manuscript needed some rewriting in order to prevent negative biases in this regard. We have processed this throughout the entire document by e.g. emphasizing differences between cultures and countries.
Comparisons of the countries in section 4 are arbitrary and based on a non-systematic literature study, yet this is the main section backing up the claims of difference. But there is no control set to compare with, resulting in diluting the strength of the conclusion.
Dear Reviewer, section 4 has now been thoroughly rewritten and we have used relevant literature to support our claims. Please see the rewritten text on Pages 9 to 13 and our new comparison text and table on page 12 and 13, lines 517-529: “Comparing the cases along the cultural dimensions using Hofstede’s model, we found that especially the difference between individualism versus collectivism is characteristic in all examples, followed by power distance, that is the perceived hierarchy and power differentials in the use of ICT, and uncertainty avoidance. The use of ICT appears to be significantly influenced, for example, by the way in which hierarchical stratifications exist in cultures that allow tools, such as e-mail, to be used as a means of communication in the hierarchy, or by the extent to which ICT is allowed to gain access to certain sources of information, or by the extent to which ICT use is permitted at all. Regarding the dimension of individualism, the question is whether in collectivistic cultures a more individually-oriented ICT application is appreciated or understood. It also seems that in order to prevent uncertainty, one adopts an increasing formalization”.
The theoretical reference of Hofstede indicators is not clear. These indicators differ a lot even for European countries. So how can it be concluded that the low diffusion of ICT in developing countries and differences of Hofstede?
Dear Reviewer, we are with you in the view that our use of the Hofstede literature was not fully clear, in particular to novice readers in this field. We have now elaborated on this issue op Page 12 (see above comment) and have added a table for the purpose of comparison on page 12, 13 (starting at line 532).
I think the main problem in this manuscript is the mix of methods and the concepts discussed. the solutions and problems are discussed in random order. I think reorganising the problems and solutions (methods) separately will improve the paper a lot. Clear research questions are needed, since the paper discusses many dimensions, such as rich/poor, communication, developed/developing regions, culture, gender, etc. which it is extremely had to systematically look into all these factors and a high level as global in a single paper.
Dear Reviewer, thank you for your conscientious reading. We have now improved the focus in our methods & concepts throughout the introduction on pages 1 - 4 and the central questions on Page 4 line 157, and we have tried our best to optimize the conceptualization of key concepts. See the following text excerpts:
Page 1,2 (lines 39-48): This contribution goes into the possibility of seeing ICT as a source of progress in cooperation among countries. In order to use ICT in this way, it is important to better understand in which way ICT and e-collaboration are perceived in developing countries? In general, ICT is seen as a key driver of economic development and as a means to reduce poverty in developing countries [96] but there are major differences in the interpretation of the term developing countries. The designations "developed" and "developing" are intended for statistical convenience and do not necessarily express a judgement about the stage reached by a particular country or area in the development process [97] These statistics include e.g. comparisons between income, economy, health, education, and safety”.
Page 2 (lines 93-95): “We posit that there is an urgent need to accompany ICT and infrastructure by human capacity, more specifically, through intercultural competencies, as there is concern about Western cultural hegemony regarding ‘the information society’ over other world regions [101]”.
Page 4 (lines 157-159): “Herewith, the central question in this study is: What is the meaning of ICT in developing countries, specifically, the importance of ‘making sense’, and can intercultural competencies help us to bridge the gap in e-collaboration between countries? “
Page 4 (lines174-181): “We will first discuss Hofstede's cultural framework [102] as a basic model for understanding cross-cultural differences after which we will look specifically at the relationship between cultural differences and ICT. This brings us to the question of the world-wide adaptation of ICT. Following this theoretical outline, we will present six characteristic case examples dealing with ICT implementation in developing countries. This limited casuistry is chosen because these non-Western cases are illustrative of the cultural determinants in the use of ICTs.”
And I see a lack of scientific rigour in the conducting of the method. If they could convert to a Systematic Literature study it might save the purpose.
Dear Reviewer, our scholarly work is not based on a systematic literature study, yet, builds on the findings of an investigation and categorization of the cultural factors that impact the adaptation and diffusion of ICT, especially in developing countries. After a thorough comparison between different intercultural competencies’ frameworks, we conclude that success in technology-mediated-collaboration is dependent on an adequate level of motivation, knowledge and skills.
Reviewer 3 Report
This empirical study summarizes and categorizes the cultural factors impacting the adaptation and diffusion of ICT in developing countries and investigates which factors hinder and/or facilitate the collaboration with Western countries. In particular, the findings of a thorough comparison between different intercultural competencies’ frameworks indicate that intercultural competencies comprise a combination of motivation, knowledge, and skills.
I would like to thank the editor to give me the opportunity to review this interesting work. The impression of the paper is interesting. However certain points require some illumination in order to fit usual academic standards.
- In the introduction part, please consider a review of the existing literature and show what is the originality of your work.
-How could/should futures studies improve the model?
-English has to be improved to overcome some mistakes.
Author Response
This empirical study summarizes and categorizes the cultural factors impacting the adaptation and diffusion of ICT in developing countries and investigates which factors hinder and/or facilitate the collaboration with Western countries. In particular, the findings of a thorough comparison between different intercultural competencies’ frameworks indicate that intercultural competencies comprise a combination of motivation, knowledge, and skills.
I would like to thank the editor to give me the opportunity to review this interesting work. The impression of the paper is interesting. However certain points require some illumination in order to fit usual academic standards.
In the introduction part, please consider a review of the existing literature and show what is the originality of your work.
Dear Reviewer, thank you very much for your time and efforts, and for the constructive feedback that has helped us a lot to improve our manuscript. And thank you very much for your compliments about our paper.
In order to add more clarity on the unique contribution of our paper to the already existing literature, we have rewritten the Introduction, have added more supporting references throughout the article and have stressed the originality of our work. See the following text excerpt:
Page 1 (lines 39-48): “This contribution goes into the possibility of seeing ICT as a source of progress in cooperation among countries. In order to use ICT in this way, it is important to better understand in which way ICT and e-collaboration are perceived in developing countries? In general, ICT is seen as a key driver of economic development and as a means to reduce poverty in developing countries [96] but there are major differences in the interpretation of the term developing countries. The designations "developed" and "developing" are intended for statistical convenience and do not necessarily express a judgement about the stage reached by a particular country or area in the development process [97] These statistics include e.g. comparisons between income, economy, health, education, and safety”.
Page 4 (lines 157-159): “Herewith, the central question in this study is: What is the meaning of ICT in developing countries, specifically, the importance of ‘making sense’, and can intercultural competencies help us to bridge the gap in e-collaboration between countries?”.
Page 14 (lines 605-616): “Intercultural competencies can be defined as “a complex of abilities that are needed to interact with people from other cultures adequately and effectively” [103] (488). Developing intercultural competencies is a solution for understanding and possibly coping with the specific cultural aspects troubling the use of ICT in developing countries. More specifically, the main issues besides ICT adaptation in different cultures comprise: exchanging knowledge, learning and e-collaboration, in our case between countries, and the rapid growth of ICT as a highly important way of communication. So far, we have mainly looked at the process of implementation of ICT in developing countries. A preliminary conclusion from this endeavor is that the success of ICT implementation is highly dependent on the underlying cultural characteristics that encompass Western ICT technology versus the meaning given to it by other cultures”.
How could/should futures studies improve the model?
In the Discussion section, we have elaborated on the recommendations for future research and have added more suggestions. See the following text excerpt:
Page 22 (lines 953-979): “Ideally, this research adopts a longitudinal approach in order to determine causal relationships between antecedents of technology-mediated collaboration, such as intercultural competencies, and its outcomes, for instance in terms of firm and societal success. It is also important to investigate whether intra-individual changes in intercultural competencies across life stages, for instances as a result of building up these competencies through more experiences, and or training and development, throughout one’s career, are related to one’s career success. We argue that intercultural competencies are highly important resources in nowadays’ labor market that that is characterized by an increasing internationalization [106].
To conclude, especially in the beginning of this scholarly project, we focused on the differences between on the one side developed countries, that are in many circumstances responsible for delivering ICT systems that are mostly developed from a Western perspective, and developing countries that make use of these ICT applications and that work from their culturally-formed perspective using these applications. However, obviously, the need for making use of intercultural competencies can come from all countries across the globe. It is not self-evident that in collaborations between the powerful west and the less powerful non-western countries both parties will show initiative to develop cultural competencies and to use them not in order to better interpret the ICT applications developed from the other culture’s perspective. For instance, in countries wherein a higher level of collectivism, power distance and uncertainty avoidance are ingrained in daily life different systemic approaches are needed in order to develop ICT infrastructure and software [107]. More research is needed in order to disentangle what determinants influence the success and failure rates of e-collaboration between countries across the globe. We hope that our contribution forms a sound basis for designing such research and that its outcomes will increase our insights in fruitful avenues for fostering technology-mediated collaboration with developing countries”.
English has to be improved to overcome some mistakes.
Dear Reviewer, an additional, conscientious proofread has been performed in order to improve the writing and to eliminate the final mistakes.
Round 2
Reviewer 2 Report
Thank you for the new version of the manuscript. I think now it has some balance of strength of the claims.
Now my major criticism is with the references. most of the references are over 10years old, which in this subject 10 years difference has a major impact. Smart technologies used widely has changed the so-called digital divide on a greater scale, whereas the internal digital divide is more problematic irrespective of which side of the globe people are coming from. ICT in developing countries is thereby kind of an old school topic in my understanding, so in order to preserve the context and the content of the paper, I would suggest backing all the strong arguments (results in the outcome of the study) to be based on publications not more than five years old.
Author Response
Dear Reviewer, thank you again very much for your time and efforts, and for the feedback that has helped us a lot to improve our manuscript.
Given the nature of our literature review, we searched for authentic sources, particularly to properly outline the developments in the adaptation process. However, we recognize the necessity of including more recent literature to better substantiate some of our statements. This has led us to include new and more recent references both for the case studies and in many places in the text.
Below we list the new used sources:
Viswanath Venkatesh, Hillol Bala, V. Sambamurthy. Implementation of an Information and Communication Technology in a Developing Country: A Multimethod Longitudinal Study in a Bank in India. Information Systems Research, 27 (3), 2016, 558-579 https://doi.org/10.1287/isre.2016.0638.
Salem, Njma et al. "A Study on the Integration of ICT by EFL Teachers in Libya". Eurasia Journal of Mathematics, Science and Technology Education, vol. 14, no. 7, 2018, pp. 2787-2801. https://doi.org/10.29333/ejmste/90594
Kultangwattana, P., Opportunity and Challenge of Human Resource Development and Saving Behavior Adaptation of Older Persons in Thai Society. Proceedings of the 2014 International Conference on Public Management. Advances in Economics, Business and Management Research. https://doi.org/10.2991/icpm-14.2014.36.
Gagliardone, I. (2014). A Country in Order: Technopolitics, Nation Building, and the Development of ICT in Ethiopia. Information Technologies & International Development, 10(1), 2014, 3–19.
Thuy, T.T.H and S. A. Qalati. Preschool teachers's attitude towards the integration of information technology into English teaching for young children in Vietnam. International Journal of Exconomics, Commerce and Management. Vol. 8, Issue 2, 2020. p 279-294.
Robin Shields. ICT or I see tea? Modernity, technology and education in Nepal, Globalisation, Societies and Education, 9:1, 2011, 85-97, DOI: 10.1080/14767724.2010.513536
Beugelsdijk, S., & Welzel, C. Dimensions and dynamics of national culture: Synthesizing Hofstede with Inglehart. Journal of Cross-Cultural Psychology, 49(10), 2018, 1469-1505.
Vollero, A., Siano, A., Palazzo, M., & Amabile, S. Hofstede's cultural dimensions and corporate social responsibility in online communication: Are they independent constructs?. Corporate Social Responsibility and Environmental Management, 27(1), 2020, 53-64.
Heimbürger, A., & Kiyoki, Y. On Temporal Aspects in Cross-Cultural e-Collaboration Between Finland and Japan Research Teams. International Journal of e-Collaboration (IJeC), 14(2), 2018, 37-54.
Linnes, C. Embracing the Challenges and Opportunities of Change Through Electronic Collaboration. International Journal of Information Communication Technologies and Human Development (IJICTHD), 12(4), 2020, 37-58.
Schulze, J., & Krumm, S. The “virtual team player” A review and initial model of knowledge, skills, abilities, and other characteristics for virtual collaboration. Organizational Psychology Review, 7(1), 2017, 66-95.
Hsu, S. Y. S., & Beasley, R. E. The effects of international email and Skype interactions on computer-mediated communication perceptions and attitudes and intercultural competence in Taiwanese students. Australasian Journal of Educational Technology, 2019, 35(1).
Wu, Z. Positioning (mis) aligned: The (un) making of intercultural asynchronous computer-mediated communication. Language Learning & Technology, 22(2), 2018, 75-94.
Riva, G. The sociocognitive psychology of computer-mediated communication: The present and future of technology-based interactions. Cyberpsychology & behavior, 5(6), 2002, 581-598.
Ao, S. H., & Huang, Q. S. A systematic review on the application of dialogue in public relations to information communication technology-based platforms: Comparing English and Chinese contexts. Public Relations Review, 46(1), 2020, 101814.
Osmani, M. W., El Haddadeh, R., Hindi, N., & Weerakkody, V. The Role of Co-Innovation Platform and E-Collaboration ICTs in Facilitating Entrepreneurial Ventures. International Journal of E-Entrepreneurship and Innovation (IJEEI), 10(2), 2020, 62-75.
Akinsola, S., & Munepapa, J. Utilisation of e-collaboration tools for effective decision-making: A developing country public sector perspective. South African Journal of Information Management, 23(1), 2021, 1-7.
Godwin-Jones, R. Telecollaboration as an approach to developing intercultural communication competence. Language Learning & Technology, 23(3), 2019, 8–28.
Sansone, N., Cesareni, D., Bortolotti, I., & Buglass, S. Teaching technology-mediated collaborative learning for trainee teachers. Technology, Pedagogy and Education, 28(3), 2019, 381-394.
Stewart, M. K. Communities of inquiry: A heuristic for designing and assessing interactive learning activities in technology-mediated FYC. Computers and Composition, 45, 2017, 67-84.
Kuyoro‘Shade, O.; Awodele, O.; Alao, O. D.; Omotunde, A. A. ICT solution to small and medium scale enterprises (SMEs) in Nigeria. International Journal of Computer and Information Technology, 2013, 4(2), 785-789.
Lamichhane, N. Implementing continual service improvement in business enterprises: A proposal to improve business effectiveness of Nepal. International Journal of Advanced Research and Publications,2019, 3, 238-251.